# Antimicrobial TiN-Ag Coatings in Leather Insole for Diabetic Foot

**DOI:** 10.3390/ma15062009

**Published:** 2022-03-08

**Authors:** Sandra M. Marques, Isabel Carvalho, Teófilo R. Leite, Mariana Henriques, Sandra Carvalho

**Affiliations:** 1CFUM-UP, Physics Department, University of Minho, 4800-058 Guimarães, Portugal; sandra.carvalho@dem.uc.pt; 2CEB, Centre of Biological Engineering, LIBRO-Laboratório de Investigação em Biofilmes Rosário Oliveira, Campus of Gualtar, University of Minho, 4710-057 Braga, Portugal; isabel.carvalho@deb.uminho.pt (I.C.); mcrh@deb.uminho.pt (M.H.); 3CEMMPRE, Department of Mechanical Engineering, University of Coimbra, 3030-788 Coimbra, Portugal; 4ICC-Indústrias e Comércio de Calçado S.A., Sol-Pinheiro, 4810-718 Guimarães, Portugal; teofilo.leite@lavoroeurope.com

**Keywords:** Ag nanoparticles, sputtering, leather, diabetic foot, antimicrobial properties

## Abstract

This work reports on TiN-Ag antimicrobial coatings deposited by d.c. magnetron sputtering on leather used for insoles on the footwear industry, studies involving the antimicrobial properties of Ag-based functionalized leathers by sputtering techniques are shown. The X-ray diffraction (XRD) results suggested the presence of crystalline fcc-TiN phase for the sample without silver, and also a fcc-Ag phase in the samples containing silver. According to the Scanning Electron Microscopy (SEM) analysis, the coatings were homogeneous and dispersed Ag clusters were detected on the surface of samples with silver content above 8 at. %. The Inductively coupled plasma—optical emission spectrometry (ICP-OES) analysis showed that the ionization of silver over time depends on the morphology of the coatings. The samples did not present cytotoxicity and only samples with incorporated silver presented antibacterial and antifungal activity, highlighting the potential of the TiN-Ag insole coatings for diseases such as diabetic foot.

## 1. Introduction

Diabetes Mellitus (DM), considered a global epidemic, is a medical disorder that affects a large proportion of the population between different age groups. One of the most serious complications of the disease is related to pathological changes in the feet, which are the most frequent cause of hospitalization in the Western world [1]. Tissue impairment worsened in the lower limbs by trauma and the vulnerability of the diabetic patient to infections, generates complex clinical conditions that are synthetically encompassed under the name “Diabetic Foot”. Thus, Diabetic Foot (DF) is the term used to designate the various injuries that can occur on the foot of the diabetic individual. This pathology translates into the appearance of skin lesions and deep planes related to neuropathic, vascular, orthopedic, infectious, and functional changes.

It is estimated that 19–34% of patients with diabetes are at risk of developing foot injuries, with ulceration secondary to progressive peripheral polyneuropathy, the most common cause of these injuries [2].

In view of this reality, several European countries, as well as the World Health Organization (WHO) and the International Diabetes Foundation (IDF), have set a target for reducing amputation rates by 50%.

According to international recommendations, the prevention of DF can be achieved through the prescription of appropriate footwear, diagnostic measures, and more effective treatments during the initial stages of the disease. These preventive measures have been shown to have positive effects on the patient’s quality of life and reduced health care costs [1,3,4,5,6]. Half of all DF ulcers occur on the plantar surface of the foot and are caused mainly by high levels of mechanical pressure that act on the foot during walking [7,8]. Thus, it has been widely accepted in the literature, that the prescription of therapeutic footwear constitutes an effective measure in the prevention of ulcers, especially for patients with peripheral neuropathy [9,10,11], since the modifications of the footwear, for this type of patient, are designed to reduce plantar pressure during walking [10].

It is known that the immunological impacts of the diabetes disease make foot ulcers highly susceptible to the prevalence of infections. In this context, the prevention of the colonization of microorganisms in the foot/wound will result in a higher healing rate, contributing significantly to the prevention of the occurrence of infection.

The aim of the present work is to develop an antimicrobial coating on the insole surface with the ability to reduce the risk of infections associated with ulcers, by depositing Physical Vapour Deposition (PVD) coatings that have silver nanoparticles (NPs) in their composition. The use of silver NPs is due to the acknowledgement of their excellent antimicrobial activity [12,13,14,15]. We intended to control the diffusion of silver in the matrix to the coating surface toto control the ionization rate of the silver into the medium and thus have a controlled release of the antimicrobial agent over time. TiN coatings will be used as matrix because they present excellent chemical stability, biocompatibility, and outstanding mechanical properties, leading to a broad range of applications, including those in the biomedical area [16,17]. The silver addition (good antimicrobial agent) to the TiN films may lead to a softer coating, since silver presents a low Young’s Modulus in-creasing the plasticity of TiN [17,18,19,20].

TiN-Ag coatings, using different current density applied to Ag target, were deposited by magnetron sputtering on leather substrates. Structural and morphological properties of the films were evaluated, together with the antimicrobial properties.

## 2. Materials and Methods

### 2.1. Coatings Preparation

TiN-Ag coatings were deposited by dc magnetron sputtering onto ultrasonically cleaned silicon substrates (monocrystalline silicon wafers (100 P-type/B) with resistivity 1–100 cm) and leather used for shoe’s interior and insoles Silicon substrates were used for Energy Dispersive Spectrometry (EDS) and X-ray Diffraction studies were performed on silicon substrates (XRD). The analyses by Scanning Electron Microscopy (SEM) were performed in silicon and in leather.

A pure Ti target (99.99%) and an Ag target (99.99%) (both 200 mm × 100 mm) were used, in Ar + N_2_ mixtures, with the substrates rotating, 70 mm away from the target, at a constant speed of 7 rpm. The base pressure in the deposition chamber was approximately 5 × 10^−4^ Pa, increasing to 1.7 × 10^−1^ Pa during deposition. Argon and reactive gas, N_2_ flow, were kept constant at 60 sccm and 3.5 sccm, respectively.

The current density applied to the Ti target was kept constant at 10 mA/cm^2^, whereas the current density applied to the Ag target was varied between 1, 1.25 and 2 mA/cm^2^. The Ti target was connected to a dc power supply, while the Ag target was connected to aa pulsed dc power supply. The frequency and reverse time were set to 200 kHz and 1536 ns, respectively, resulting in a 69% duty cycle.

Substrates were subjected to plasma treated before coating deposition to improve adhesion, with Ar flow of 80 sccm for 30 min. A pulsed dc power supply was applied to the substrate-holder with a pulse width of 1536 ns, frequency of 200 kHz, and intensity of 400 mA.

In order to avoid structural damage of the leather substrate, the temperature must be ideally kept below 130 °C even if in the range of 130–170 °C for long periods itit does not induce structural changes [21]. In this sense, the depositions were performed at 100 °C with no bias polarization applied to the substrate-holder. The deposition time was maintained constant, at 1800 s.

### 2.2. Chemical and Physical Analyses

Chemical composition estimation was performed with an EDAX-Pegasus X4M-Energy dispersive spectrometer (EDS) apparatus coupled with a SEM. The structure and phase distribution of the coatings were analyzed by XRD using a D8 Discover diffractometer, Bruker, Massachusetts, USA (Cu K radiation–λ = 1.5406 Å 5406 Å, step 0.04°, time per step 1 s, and 6–60 2θ2θ interval). The surface morphology was examined by SEM through a NanoSEM–FEI Nova 200, FEI Company, Hillsboro, OR 97124-5793 USA (FEG/SEM) equipped with a field emission gun (FEG), operated in high vacuum mode with a chamber pressure of 0.003 Pa. The micrographs were obtained with secondary (SE) electron detectors in “through-the-lens” (TLD) mode at an acceleration voltage of 10 kV and working distance of roughly 5 mm. Measurements were performed in three areas randomly chosen in all samples before and after the biological tests. A magnification of 5000× and 50,000× was used.

The wettability of the different samples was determined using a contact angle meter apparatus (OCA 15 Plus, DataPhysics Instruments GmbH Filderstadt, Germany). All measurements were performed at room temperature and two microliters drops of pure water (polar liquid); glycerol (polar liquid, Sigma, Merck KGaA, Darmstadt, Germany) and α-bromonaphthalene (apolar liquid, Sigma, Merck KGaA, Darmstadt, Germany), were used as reference liquids. Using the Owens, Wendt, Rabel, and Kälble (OWRK) method, the surface free energy was calculated [22].

An inductively coupled plasma optical emission spectrometer (ICPOES–ICP PerkinElmer spectrometer model Optima 8000) was used to measure silver ion release. Uncoated and coated leathers were submerged in a vessel containing 20 mL of the synthetic sweat (S.S.) solution (the S.S. solution preparation according to ISO 105-E04:2013) at room temperature. After 2, 6, 12, 24, 48, 120 and 168 h of immersion, a volume of 2 mL of the solutions was removed. After dilution in 4 mL of HNO_3_ to dissolve any remaining chemicals present in solution, the solutions were filtered over a 0.2 µm cellulose membrane and stored in the dark to prevent precipitation for ICP-OES. At least 3 samples were evaluated, and the standard deviation are presented. The results are expressed in ng of silver release in one ml of solution by 1 cm^2^ of sample (ppb/cm^2^).

### 2.3. Biological Analyses

#### 2.3.1. Cytotoxicity Assay

Cytotoxicity tests were performed using 3T3 fibroblasts (CCL-163) obtained from the American Type Cell Collection.

The cytotoxicity was tested by an indirect test performed on the S.S. solution which was in contact with TiN-Ag samples (previously sterilized at UV light during 1 h) for 24 h, in accordance with ISO 10993-5:2009.

Cells were grown in Dulbecco’s Modified Eagle’s Medium (DMEM, Biotecnómica, São Mamede Infesta, Portugal) supplemented with 10% of fetal bovine serum (FBS, Biotecnómica, São Mamede Infesta, Portugal) and 1% zellshield (Biotecnómica, São Mamede Infesta, Portugal) and, after attaining 80% of confluence, cells were detached and 50 μL of cell suspension containing 1 × 10^5^ cells/mL was added to each well of a 96 wells’ plate (Orange Scientific, USA). Then the plates were incubated with 5% CO_2_ at 37 °C for 24 h. Afterwards, 50 μL of S.S. solution that was in contact with samples were added to cells and they were again incubated under the same conditions for further 24 h. After that time, 20 μL of MTS (3-(4,5-dimethylthiazol-2-yl)-5-(3-carboxymethoxyphenyl)-2-(4-sulfophenyl)-2H-tetrazolium), inner salt (Promega CellTiter 96^®^ AQueous Non-Radioactive Cell Proliferation Assay) was added to each well, filled and incubated in the dark. After 1 h, the absorbance of the resultant solution was read at 490 nm. The percentage of cellular viability was determined using the following expression:(1)$$Viab\%=\;\frac{{\left[OD\right]}_{490S}}{{\left[OD\right]}_{490C}}\;\times 100$$
where$\;{\left[OD\right]}_{490S}$ means the measured value optical density of sample (cells’ growth in the presence of samples (S. S. solution that was in contact with TiN-Ag coating leathers)) and$\;{\left[OD\right]}_{490C}$ means the measured value optical density of control (cells’ growth in the absence of samples). The assays were carried out at least three times and in triplicate.

#### 2.3.2. Antimicrobial Activity

##### Agar Diffusion Test

The antimicrobial activity of the TiN-Ag coatings deposited on the leather, were tested against four different microorganisms: two fungal species *Candida parapsilosis* (ATCC 22019, a clinical isolate belonging to the CEB Biofilm Group collection) and *Trichophyton mentagrophytes* (MUM08.08T. Mentagrophytes); and two bacteria’s, one gram-positive, *Staphylococcus aureus* (ATCC 6538, a clinical isolate belonging to the CEB Biofilm Group collection) and one gram-negative, *Escherichia coli* (CECT 434, a clinical isolate belonging to the CEB Biofilm Group collection).

*S. aureus*, *E. coli* and *C. parapsilosis* were cultivated in liquid medium, by inoculation of a single colony in 30 mL of Tryptic Soy Broth (TSB, Frilabo) for bacteria and Sabouraud Dextrose Broth (SDB, Frilabo) for the yeast, then microorganisms were incubated for 18 h at 37 °C and 120 rpm. Afterwards, the resultant cell suspension was adjusted to an optical density (OD) of 1.0 at 620 nm for bacteria, and properly diluted in TSB to 1 × 10^7^ CFU mL^−1^. For *C. parapsilosis*, the cell density was further adjusted to 1 × 10^7^ cells/mL, using a Neubauer hemocytometer (Marienfeld, Lauda-Königshofen, Germany). To the incubation of the microorganisms in the agar, an aliquot of cell suspension (100 µL) was dispersed on TSA or SDA petri dishes. Then, the samples (previously sterilized by exposure of ±1 h to UV light) were placed separately on the top of an agar plate, with the coated side in contact with the agar, and incubated for 24 h at 37 °C.

To evaluate the antifungal activity, a method based on ISO 20645/2005 was adapted. *T. mentagrophytes* was seeded on 2 mL of sloping potato dextrose agar (PDA, Frilabo) at room temperature for 7 days. Afterward, 3 mL of distilled water was added and shaken, and the cell density was adjusted to the optical density (OD) of 0.08–0.1 at 620 nm and properly diluted to 1 × 10^5^ CFU mL^−1^. The coatings were placed on two-layer agar plates.

The lower layer consists of culture medium potato dextrose agar (PDA, Frilabo) free from fungi and the upper layer is inoculated with the fungus. For the lower layer, 10 ± 0.1 mL PDA were poured into sterile petri plates. After sterilization control, for the upper layer, molten PDA (precooled to approximately 45 °C) was inoculated with fungal culture (1 × 10^7^ CFU mL^−1^), vessel was shaken vigorously to distribute fungi evenly, then the plates were incubated at room temperature for approximately 7 days.

After the incubation period, the halo (zone of transparent medium, which means that there is no microorganisms growth) formed around the sample was photographed to record the results (pictures acquired using Image LabTM software). All experiments were carried out in triplicate per sample and repeated in at least three independent assays. Zone of inhibition (ZoI) tests, adapted from Kirby–Bauer test [23], were carried out to determine the diffusion of silver from the coating surface.

Scanning electron microscopy (SEM) was used to observe the coatings surface after halo test and three fields were used for image analysis.

##### Biofilm Formation

In addition, bacterial colonization assays were performed. *S. aureus* and *E. coli* and *C. parapsilosis* were cultivated in liquid medium, by inoculation of a single colony in 30 mL of Tryptic Soy Broth (TSB, Frilabo) for bacteria and Sabouraud Dextrose Broth (SDB, Frilabo) for the yeast, then microorganisms were incubated for 18 h at 37 °C and 120 rpm. For *C. parapsilosis*, the cell density was further adjusted to 1 × 10^7^ cells/mL, using a Neubauer hemocytometer (Marienfeld, Lauda-Königshofen, Germany) (this procedure is the same above mentioned, see Agar Diffusion Test Section 1).

Afterwards, the resultant cell suspension was adjusted to an optical density (OD) of 1.0 at 620 nm and properly diluted to a final concentration of approximately 1 × 10^7^ CFU mL^−1^ (this concentration is required in order to maintain the same order of magnitude as the agar diffusion test, see Agar Diffusion Test Section). Coated and uncoated coupons, with 10 mm of diameter (previously sterilized by exposure of ±1 h to UV light) were inserted in 24-well plates and 1 mL of cellular suspension was added to each well. The plates were then incubated at 37 °C under 120 rpm for 24 h. After incubation, the coatings were gently washed with Phosphate Buffered Saline (PBS (1×)) to remove non-attached bacteria. Thereafter the coated coupons were transferred to new 24-well plates and 1 mL of PBS (1×) was added, the entire surface of the coupons was properly scraped to detach the formed biofilm. For the extraction of the biofilm matrix, the resulting suspension together with the coupon were transferred to an eppendorf and sonicated for 30 s to 30%. Finally, a serial of dilutions on TSA plates were incubated at 37 °C for 24 h and, proceed with the counting of the number of Colony forming units (CFU). All assays were independently performed in at least three independent assays.

Results from biological assays were compared using one-way analysis of variance by applying the Bonferroni multiple comparisons test, using the software GraphPad Prism. All tests were performed with a confidence level of 95%.

## 3. Results and Discussion

### 3.1. Chemical Composition vs. Deposition Parameters

The synthesis conditions together with the coatings chemical composition, thickness and deposition rate, are summarized in Table 1.

Through the deposition time and coating thickness, estimated by SEM cross-sectional imaging on a silicon substrate, it was possible to determine the deposition rate. All the coatings were deposited with a constant power density applied to titanium (J_Ti_) target of 10 mA/cm^2^ and increasing the power density applied to silver (J_Ag_) target from 1, 1.25, and 2 mA/cm^2^, respectively, while the N_2_ flow was remain constant at 3.5 sccm, to achieve different silver content. The presence of oxygen and carbon in sputtered coatings occurs as result of contamination, thus, the oxygen and carbon content were also determined in EDS analysis being found that its amount is around 20 at. %. TiN sample presents a N/Ti ratio close to 1, which corresponds to the formation of a stoichiometric film, as achieved in previous works [19,24]. The deposition rate increased with the increase of the power density applied to silver target, which was expected since Ag presents a higher sputtering yield than Ti (3.12 for Ag and 0.51 for Ti, when bombarded with Ar at 0.5 keV [25,26]). The Ag content is influenced by the deposition conditions: the increase in J_Ag_ leads to an increase in Ag content. The relation between current density and sputter efficiency is well described in [27] and already discussed by the authors in previous studies [19,26].

### 3.2. Morphology

SEM top-view and cross-section analyses were performed to evaluate surface characteristics and the presence of Ag nanoparticles on the surface, that would be available for a fast-antimicrobial activity. The SEM micrographs of representative coatings are depicted in Figure 1.

According to the results obtained from SEM analysis (Figure 1), the TiN sample is homogeneous presenting a very smooth surface, while the coatings with silver are composed by nanoparticles embedded in a TiN matrix. Several authors [19,24,28], have reported that Ag-TiN_x_ coatings were composed by Ag clusters segregated from the TiN grain boundaries, which appear as bright spots in the SEM micrographs. Silver clusters can diffuse through the base coating, resulting in a non-uniform Ag distribution over the coating thickness. To elucidate these conclusions, the morphology of the coatings was analyzed by cross-sectional SEM micrographs of the different samples. These cross-section micrographs show a columnar-type structure common in this type of films with relative low deposition temperature according to the Thornton diagram [29]. All depositions (mainly for highest silver contents) present nanoparticles in the boundaries of the columns quite well distributed along the coating thickness. In samples TiN-Ag1 and TiN-Ag1.25 silver clusters can be found on the surface coatings.

The surface hydrophobicity of the different samples was assessed by contact angle measurements, presented in Figure 2. The surface energy of the various coatings were calculated according to the Owens, Wendt, Rabel, and Kälble (OWRK) approach [22], which is a standard procedure for calculating the surface free energy of hydrophobic solid surfaces from the contact angle using at least two liquids (Table 2).

Surfaces can be classified as super-hydrophilic when, in the case of liquid water, θ ≈ 0°, hydrophilic when θ < 90° and hydrophobic when θ > 90° [29]. For all samples, TiN, TiN-Ag1, TiN-Ag1.25 and TiN-Ag2 (Figure 2), θ values are higher than 90°, suggesting a coating hydrophobic character [16,30,31,32]. Looking to the surface energy of the different samples (Table 2) and splitting in the two components, polar and dispersive, it is possible to observe that the dispersive component presents higher values comparing with the polar component of the surface energy.

These results are aligned with the hydrophobic character observed on the contact angles of the three liquids used.

### 3.3. Structural Analysis

XRD analysis was performed to identify the phase composition in TiN-Ag coatings with the increase of silver content. The coatings XRD patterns are depicted in Figure 3. The reference peaks of the main crystalline phases are identified namely fcc-Ag (ICDD 00-004-0783), stoichiometric fcc-TiN (ICDD 00-038-1420) and hexagonal Ti (ICDD 00-044-1294).

The differences in chemical composition correlate well with the differences observed in the developed structure. The XRD pattern of TiN coating indicates that the most intense peaks are located at 36.8° and 42.9°, which are very close to the TiN (111) and TiN (200) peaks. Previous studies in TiN-Ag coatings deposited by magnetron sputtering reported this behavior [19,24].

Regarding the coatings with silver, TiN-Ag1 and TiN-Ag1.25, the presence of three crystalline phases can be clearly identified: namely a stoichiometric TiN, an Ag phase and a hexagonal Ti phase. The ratio of Ti/N increase with the increase of the power density applied to silver target, showing an excess of Ti comparing with the nitrogen, which explains the presence of the Ti phase.

For the sample with more silver, TiN-Ag2, the peaks of the TiN phase are less intense, and the peaks of the Ag phase are more intense. These are in agreement with the differences observed in the developed structure (see Figure 1), where in TiN-Ag2 sample the morphology changes and no Ag clusters are observed on the surface of the coating.

### 3.4. Silver Ions Release

The results of Ag ion release from the surface of uncoated and coated leathers immersed in S.S solution at room temperature for 7 days are presented in Figure 4.

The amounts of silver ions released from uncoated leathers were below the detection limit (~5–6 ppb) as expected due toto the absence of this element in these materials. The ionization of silver by the sample with the lowest Ag content (TiN-Ag1) over time was always lower in relation to the samples with higher contents, reaching a maximum of about 98 ppb/cm^2^ after 7 days.

Samples with higher silver content (TiN-Ag1.25 and TiN-Ag2) show a different behaviour, since the sample with intermediate silver content (14 at. %) presents a higher silver ionization than TiN-Ag2 (31 at. %) until the fifth day (180 and 114 ppb/cm^2^, respectively). This result corroborates the morphology of the samples, as previously mentioned (Section 3.2 Morphology), TiN-Ag1.25 show silver agglomerates on its surface (Figure 1), contrary to the TiN-Ag2 sample, where silver is incorporated in the coating bulk, delaying the Ag ionization during the first days of immersion.

### 3.5. Cytotoxicity Assay

Cytotoxicity results are shown in Figure 5. The non-cytotoxicity of the coated leathers was confirmed by the percentage of viable cells above 70% (88, 87 and 82% to TiN-Ag1, TiN-Ag1.25 and TiN-Ag2, respectively), according to ISO 10993-5:2009. Thus, showing that the coatings in studies are suitable to be applied to insoles.

### 3.6. Antimicrobial Properties

Microorganisms such as bacteria and fungi are frequently observed on leather products and can cause skin diseases, undesired odors, or discoloration. Leather typically presents a hydrophilic character, so it is a good medium for microbiological growth. Finishes and antimicrobial coatings on leather surface can help to prevent and manage cross-infection as well as it may extend the lifetime of the product, by inhibiting microorganism growth. Silver ions or silver nanoparticles have been known to possess strong inhibitory and antimicrobial effects in a broad microorganism spectrum [33,34,35].

The antimicrobial activities were evaluated qualitatively (inhibition zone) using the agar plate diffusion method (Figure 6) and quantitatively, by the number of viable cells (CFU) (Figure 7).

The results of agar diffusion tests are shown in Figure 6. All samples with silver present a certain inhibiting effect on the growth of *C. parapsilosis*, *T. mentgrophytes*, *S. aureus* and *E. coli*, because the growth of an inhibition halo is clear (transparent biological medium, with no bacteria growth) formed around the samples. Furthermore, Table 3 shows the values of the inhibition zone diameter of the different microorganisms for the samples with silver and it can be observed a slight increase in halo diameter with increasing silver content of the samples. These results are in agreement with the chemical composition of the samples presented on Table 1.

Through the SEM images obtained after the halo test, it is possible to see the absence of bacteria and fungi on the surface of the samples. In the sample with higher amount of silver (TiN-Ag2), the fungi cells (*C. parapsilosis*) are surrounded by silver particles thus inhibiting its activity and in the TiN sample, without silver, the surfaces show a vast microbial colonization.

To assess the number of viable cells, the cellular concentration was determined by CFU (Figure 7) in terms of the logarithm of bacterial concentration (CFU mL^−1^).

Figure 7 shows that the presence of silver on the TiN-Ag coatings promotes a reduction in biofilm formation (this reduction is statistically significant (*p* < 0.0001) and in all cases, the value is higher than 2 log, according to the Japanese Industrial Standard Z 2801:2000, which stated those values greater than 2 log means antimicrobial activity). These assays confirm the antimicrobial activity of TiN-Ag coatings. Conversely, TiN coatings did not show any reduction in biofilm formation, thus suggesting the lack of antimicrobial activity, in good agreement with the halo tests (Figure 6). Although TiN-Ag2 presents much more silver than TiN-Ag1.25 and TiN-Ag1 (31 at. %, 14 at. % and 8 at. %), respectively), thus not presents a larger halo. These results are in line with the number of viable cells determined by CFU (Figure 7), the three samples with silver (TiN-Ag1, TiN.Ag1.25 and TiN-Ag2), present a similar behavior. These results are also compatible with the Ag^+^ ions release, since after 24 h, the release of these ions was lower for the sample with the lowest amount of Ag (TiN-Ag1) (~40 ppb/cm^2^), as expected (Figure 4). However, in relation to the samples with higher Ag content (TiN-Ag1.25 and TiN-Ag2), the TiN-Ag1.25 sample with 14 at. % of silver released more Ag ions (~150 ppb/cm^2^) than TiN-Ag2 sample with 31 at. % of silver (~84 ppb/cm^2^) (behavior already explained in the previous Section 3.4. Silver ions release). This performance shows that silver ions released cannot be the main antibacterial mechanism, as previously reported in [36]. In this sense, it can be concluded the Ag^+^ ions release out of the surface into the agar appears to be independent on the form in which silver is present (segregation as clusters or nanoparticles at the grain boundaries). The authors [17], already reported antimicrobial activity with the formation of an inhibition zone around TiN-Ag coatings with Ag contents between 7 and 10 at. %. In addition, Kelly et al. [37] observed the antimicrobial activity of Ag-doped ZrN and CrN coatings using the agar diffusion test.

All these results indicate that the Ag content itself does not ensure the antimicrobial efficiency, but guarantees that the structure, phase composition and the form of Ag segregation play an important role.

## 4. Conclusions

TiN and TiN-Ag thin films were successfully deposited on leather substrates by magnetron sputtering without damaging the substrate. All samples present a crystalline fcc-TiN phases, combined with crystalline fcc-Ag phases for samples deposited with silver. According to the results, the coatings are homogeneous, presenting a TiN matrix with Ag clusters visible on the surface, for the samples deposited with silver. Contact angle measurements revealed the hydrophobic nature of the film’s surface with water contact angles higher than 90° and higher values of dispersive component compared to the polar component of the surface energy.

It has been demonstrated that all samples deposited with silver shows antibacterial and antifungal activity. From the ICP-OES analysis, it can be concluded that the ionization of silver over time depends on the morphology of the coatings Consequently, it is possible to conclude that the Ag content itself does not ensure the antibacterial and antifungal efficiency, but the structure, phase composition and the form of Ag segregation play an important role.

Further studies should be carried out to clarify the effect of silver and/or silver ion release. This work highlighted the potential of TiN-Ag coatings, and it was an essential step towards the assessment of the ideal system acting as antimicrobial coating on the insole surface with the ability to reduce the risk of infections associated with diabetic foot.

## Figures and Tables

**Figure 1 materials-15-02009-f001:**
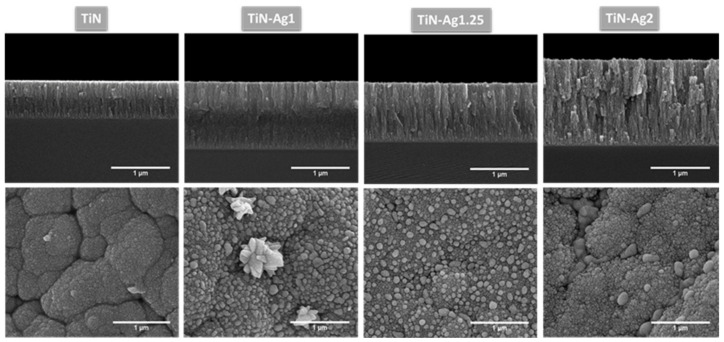
SEM micrographs on SE mode of different TiN-Ag thin films deposited on leather and silicon substrates with 100,000× magnification. Images on top row are micrographs of cross-section of the different coatings deposited on Si substrates. In the bottom row are micrographs of the surface of the different coatings deposited on leather substrates.

**Figure 2 materials-15-02009-f002:**
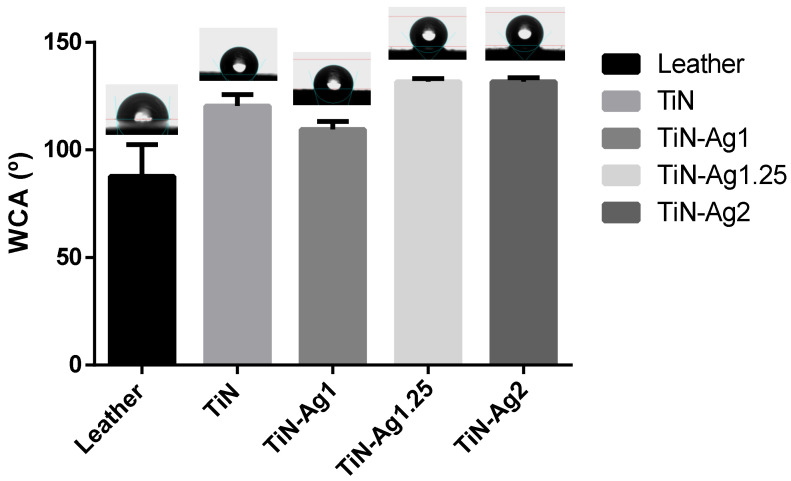
The water contact angle of the several samples, represented as a mean ± standard deviation.

**Figure 3 materials-15-02009-f003:**
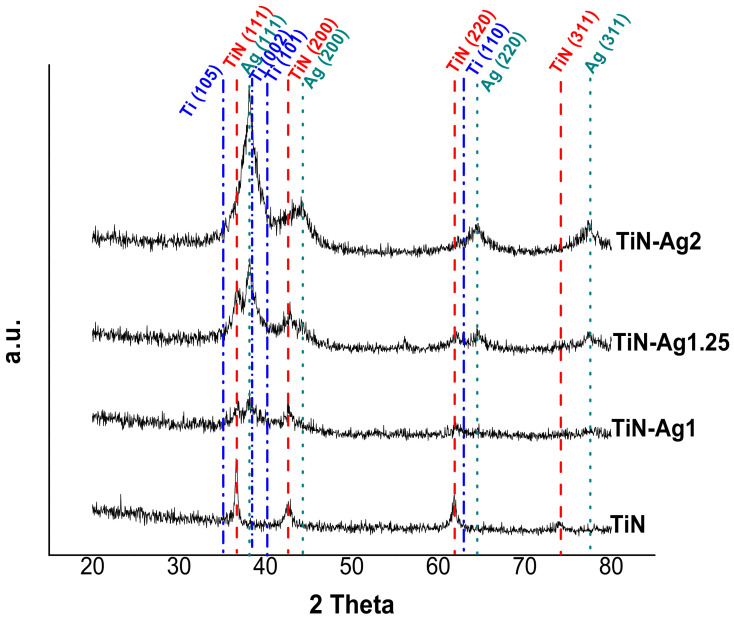
XRD patterns of TiN-Ag coatings deposited with different silver content (Cu Kα radiation).

**Figure 4 materials-15-02009-f004:**
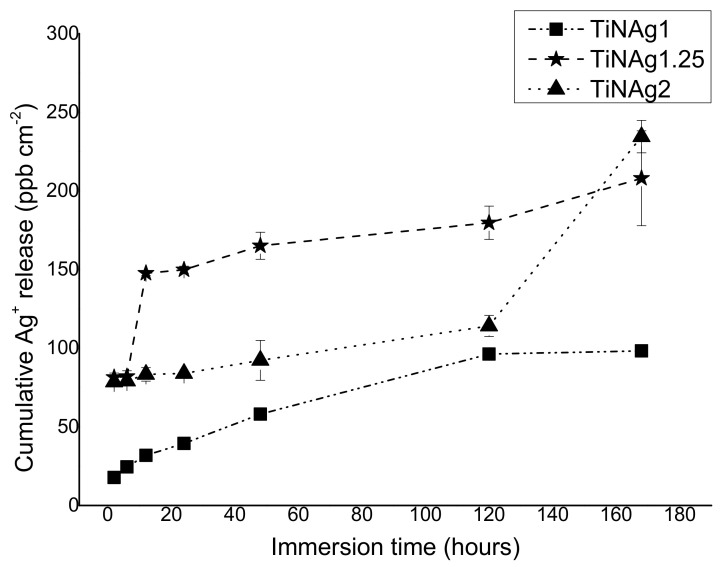
Ag^+^ release along time (2 h, 6 h, 12 h, 24 h, 48 h, 120 h and 168 h) determined by ICP-OES analysis.

**Figure 5 materials-15-02009-f005:**
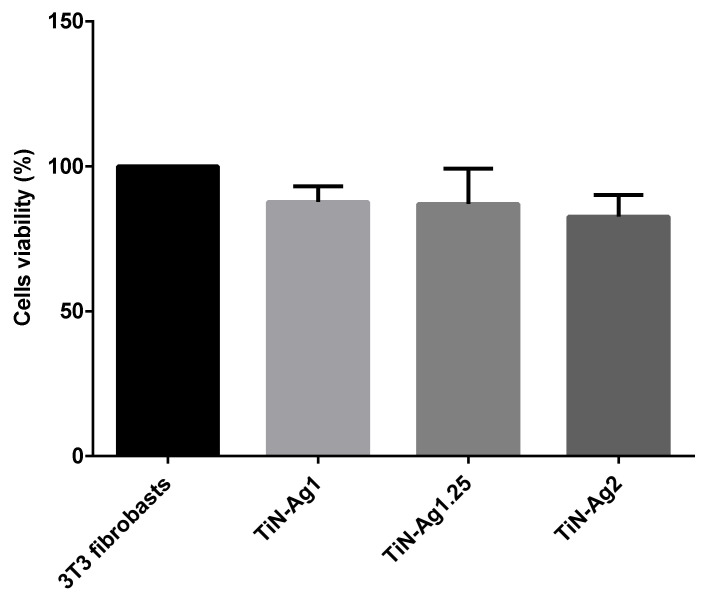
Cell viability evaluated by the MTS assay to S.S. solution after 24 h of contact with the respective samples. Fibroblasts 3T3 are the control.

**Figure 6 materials-15-02009-f006:**
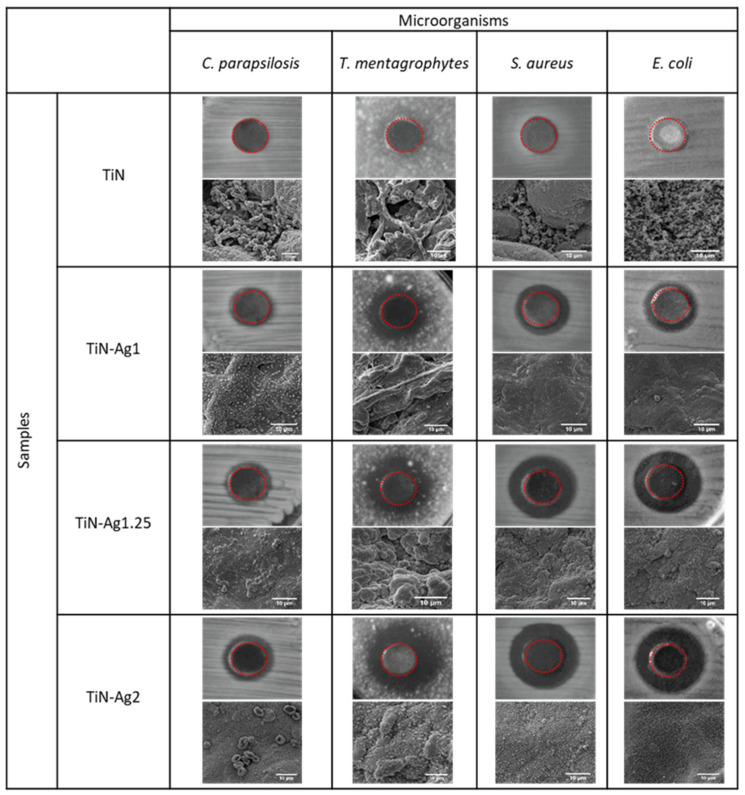
Antimicrobial activity of different samples: TiN, TiN-Ag1, TiN-Ag1.25 and TiN-Ag2 against *C. parapsilosis*, *T. mentgrophytes*, *S. aureus* and *E. coli*, evaluated by zone of inhibition assays and SEM images of the samples after the halo test. Red dashed circles highlight samples.

**Figure 7 materials-15-02009-f007:**
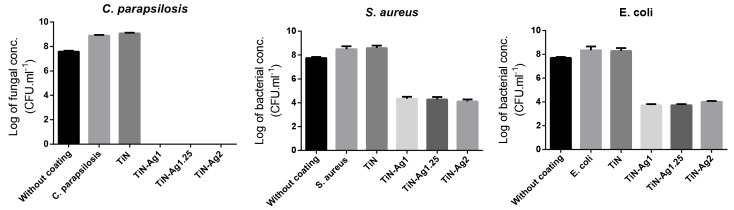
Logarithm of microbial concentration after 24 h contact between *C. parapsilosis*, *S. aureus* and *E. coli* and the different samples: TiN, TiN-Ag1, TiN-Ag1.25 and TiN-Ag2.

**Table 1 materials-15-02009-t001:** The chemical composition, as well as some experimental details. The power density applied to the Ag target was used to identify the Ag-TiNx coatings.

Coating	J_Ag_ (mA/cm^2^)	J_Ag_/J_Ti_	Thickness (µm)	Deposition Rate (µm/h)	Chemical Composition (at. %)					Ti/N	Ag/Ti
					Ti	N	Ag	C	O		
TiN	-	-	0.6	1.2	30	23	-	25	22	1.3	-
TiN-Ag1	1	0.1	1.2	2.4	28	18	8	22	24	1.6	0.3
TiN-Ag1.25	1.25	0.13	1.0	2.0	27	14	14	21	24	1.9	0.5
TiN-Ag2	2	0.2	1.5	3.0	22	12	31	17	18	1.8	1.4

**Table 2 materials-15-02009-t002:** Water (θ_W_), glycerol (θ_G_) and αααα-bromonaphthalene (θ_B_) contact angles, surface energy components: polar component (γ^p^), dispersive component (γ^d^) and total surface energy component (γ^Tot^) of the different samples.

Samples	Contact Angle ± SDA (deg)			Surface Energy Components (mN/m)		
	θ_W_	θ_G_	θ_B_	γ^p^	γ^d^	γ^Tot^
Leather	88 ± 2	87 ± 2	22 ± 6	0.3 ± 0.2	42 ± 2	42 ± 2
TiN	121 ± 5	77 ± 9	12 ± 5	1 ± 1	44 ± 1	45 ± 1
TiN-Ag1	110 ± 4	66 ± 3	15 ± 5	1 ± 0.4	43 ± 1	45 ± 1
TiN-Ag1.25	132 ± 1	89 ± 3	26 ± 10	4 ± 1	43 ± 3	47 ± 3
TiN-Ag2	131 ± 2	82 ± 10	14 ± 5	5 ± 1	43 ± 1	49 ± 1

**Table 3 materials-15-02009-t003:** Zone of inhibition diameter.

Samples	Inhibition Zone Diameter (mm)			
	*C. parapsilosis*	*T. mentagrophytes*	*S. aureus*	*E. coli*
TiN-Ag1	11 ± 0.5	18 ± 0.5	14 ± 0.2	15 ± 0.2
TiN-Ag1.25	13 ± 0.3	19 ± 0.9	17 ± 0.1	19 ± 0.2
TiN-Ag2	14 ± 0.4	22 ± 0.1	18 ± 0.1	20 ± 0.6

## Data Availability

The data presented in this study are available on request from the corresponding author.

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
