# Peer review of "Antimicrobial TiN-Ag Coatings in Leather Insole for Diabetic Foot"

_materials, 2022, doi:10.3390/ma15062009_

Round 1

Reviewer 1 Report

In this manuscript TiN-Ag films were deposited on leather for treating diabetic foot. It extended the applications of silver-containing hard coatings. I suggest the following comments for revision of the manuscript.

-The introduction could be more concise, for the work is focused on materials.

-Line 124, and so on, what is S.S solution.

-L147, why four leather types.

-‘2.3.2. Antimicrobial Activity 2.3.3. Biofilm Formation’. I don’t agree the titles. Both tests belong to Antimicrobial test. L314, plate diffusion or agar diffusion (Line 320) may be better? 2.3.3 described the normal plate counting method of Antimicrobial test. Biofilm means bacteria grown within the film composed of exopolymeric substances.

-Line 200, Using a scraper, the adher, please specify the tool and the process.

-Line 226, Ag content increases when the JAg/JTi ratio decreases, whereas Ti content de-, wrong.

-Line 237, whereas the silver coatings are… there is no silver coating

-Line 259, water, hydrophilic when they are 90°, an, wrong

Line 287, The Ti/N ratio rises as silver concentration rises, not accurate

L316, the rings are not clear. Pls add the bars, the samples and the edges of ZOI and plates (?), together with the ring diameters (data).

-Line 355, ‘In this context, we may infer that the release of Ag+ ions from the surface into’, it was reported the silver particles size may influence the ion release, which can be measured.

-There are details to be corrected, e.g. ‘amples with silver content The samples’, N2 (Line 89), 100L (L161), coat-ings, 36.8o and 42.9o, 1-bromonaphthalene and α-bromonaphthalene, ‘he yeast, Microorganisms w’, bacterial concentration (CFU ml1), statistically significant (P 0.0001), Also Kelly et al. [37] also us, etc.

-Some references need correction.

Reviewer 2 Report

The following points should be considered before publishing.

  1. Please provide the full form of PVD.
  2. Please clearly mention the novelty of the work in the abstract section.
  3. Please check thoroughly the entire manuscript. Many English corrections are needed. For example,

Line no. 16: please correct to “…and only the samples with…..“

Line no. 15:  Please add “.” after the end of the previous sentence.

Line no. 71: Please correct “a efficient…” to “an efficient..”.

Etc.

  1. Fig 1 caption: please explain what the upper and lower rows of photos are.
  2. Where is Figure 4? Please check.
  3. It looks like too many paragraphs in the conclusion part. Please check if possible to minimize the number of paragraphs.
  4. Please explain in the conclusion section why the hydrophobic character of the film surfaces was observed.

Round 2

Reviewer 1 Report

The present version has included the ions concentration result and other improvements. There are still problems in grammar and writing could be improved. Fig 7 is not at the right place and covers the text.  The forms are still consistant for bromonaphthalene . No revision and response on 'Biofilm Formation' that the reviewer had indicated. I would suggest further revision of the manuscript. 

Author Response

We would like to thank the reviewer for the excellent comments and suggestions that certainly helped to improve the manuscript. We believe that the present version meets the requirements for publications. Our replies to the referee comments are in bold and the corresponding changes in the paper are marked using the “Track Changes” function of MS Word.

Mariana Marques on behalf of authors

Referee #1 Comments to Author:

The present version has included the ions concentration result and other improvements. There are still problems in grammar and writing that could be improved.

The authors apologize but in fact, figure 4 was wrong, and we have corrected grammatical and spelling errors.

Fig 7 is not at the right place and covers the text. 

Fig. 7 is now in the right place. Sorry, was a problem with the document formatting.

The forms are still consistant for bromonaphthalene.

Corrected to α- bromonaphthalene in the document.

No revision and response on 'Biofilm Formation' that the reviewer had indicated.

The authors apologize but did not realize that the reviewer wanted the full description, in fact, the biofilm is a complex structure composed of exopolymeric substances that form the matrix of the biofilm. In this sense, we completed the information in the manuscript with the following paragraph for better understanding:

“Coated and uncoated coupons, with 10 mm of diameter (previously sterilized by exposure of ±1 h to UV light) were inserted in 24-well plates and 1 ml of cellular suspension was added to each well. The plates were then incubated at 37 °C under 120 rpm for 24 hours. After incubation, the coatings were gently washed with Phosphate Buffered Saline (PBS (1x)) to remove non-attached bacteria. Thereafter the coated coupons were transferred to new 24-well plates and 1 ml of PBS (1x) was added, the entire surface of the coupons was properly scraped to detach the formed biofilm. For the extraction of the biofilm matrix, the resulting suspension together with the coupon was transferred to an eppendorf and sonicated for 30 s to 30%. Finally, a series of dilutions on TSA plates were incubated at 37 °C for 24 h and, proceed with the counting of the number of colony forming units (CFU). All assays were independently performed in at least three independent assays”

 I would suggest further revision of the manuscript.

The manuscript was revised again.
